# Radiocontrast Media Hypersensitivity Reactions in Children

**DOI:** 10.3390/medicina58040517

**Published:** 2022-04-05

**Authors:** Francesca Saretta, Silvia Caimmi, Francesca Mori, Annamaria Bianchi, Paolo Bottau, Giuseppe Crisafulli, Fabrizio Franceschini, Lucia Liotti, Claudia Paglialunga, Giampaolo Ricci, Carlo Caffarelli

**Affiliations:** 1SC Pediatria, Ospedale Latisana-Palmanova, Dipartimento Materno-Infantile Azienda Sanitaria, Universitaria Friuli Centrale, 33100 Udine, Italy; francescasaretta@gmail.com; 2UOC Pediatria Fondazione IRCSS Policlinico San Matteo, 27100 Pavia, Italy; silviamaria.caimmi@unipv.it; 3Unità di Allergologia, Dipartmento di Pediatria, Ospedale Anna Meyer, 50132 Firenze, Italy; francesca.mori@meyer.it; 4UOC Pediatria, Ospedale San Camillo Forlanini, 00152 Roma, Italy; annamaria.bianchi9@gmail.com; 5Dipartimento di Pediatria e Neonatologia, Ospedale di Imola (BO), 40026 Imola, Italy; p.bottau@gmail.com; 6UOC di Pediatria, Dipartimento Materno-Infantile, Università di Messina, 98100 Messina, Italy; crisafullig@unime.it; 7UOC Pediatria, Azienda Ospedaliero-Universitaria “Ospedali Riuniti”, 60123 Ancona, Italy; allped@libero.it (F.F.); lucialiotti@libero.it (L.L.); 8UOC di Pediatria, Azienda Ospedaliera-Universitaria Consorziale “Policlinico-Ospedale Pediatrico Giovanni XXIII”, 70120 Bari, Italy; clapag07@gmail.com; 9Department of Medical and Surgical Sciences (DIMEC), University of Bologna, 40138 Bologna, Italy; giampaolo.ricci@unibo.it; 10Clinica Pediatrica, Dipartimento Medicina e Chirurgia, Università di Parma, 43126 Parma, Italy

**Keywords:** radiocontrast media, hypersensitivity reaction, children, IgE, skin prick test, intradermal test, iodine, allergy, drug provocation test, tryptase

## Abstract

Hypersensitivity reactions to radiocontrast media seem to be rare in children. Furthermore, the use of radiocontrast media in children remains quite safe in terms of the severity of reactions. Since pediatric guidelines are lacking, the diagnostic workup employed in adults could be adapted to children, taking into account that results have not yet been validated in this age group. Specific protocols for risk stratification and management of severe reactions have been proposed so far.

## 1. Introduction

An increased frequency of hypersensitivity reactions (HRs) to radiocontrast media (RCM) has been reported in the last decade, probably due to widespread use. However, HRs to RCM are rare in children, and it is difficult to correctly estimate their incidence due to the lack of controlled studies limited to this age group [1]. Prospective studies are difficult to manage because of the rarity of the condition in childhood. Furthermore, in many studies, differences between physiologic and allergic reactions have not yet been specified, with great heterogeneity in the definition of “severity”. Pediatric guidelines on RCM HRs are lacking. This review will focus on the current knowledge of RCM HRs in childhood. We conducted a literature search using the PubMed database, using the terms “radiocontrast media” AND “hypersensitivity” AND “children”, in order to select all the most relevant papers to discuss in this review. The search was performed for publications ranging between 2010 and 2022, limiting articles to humans and the English language. According to the above-mentioned inclusion criteria, potentially eligible publications were screened and analyzed, and no relevant publications were excluded. This review provides clinicians with an overview of the prevalence, clinical features, and pathophysiology of RCM HRs in children and summarizes all recommended diagnostic and management approaches. Most of our recommendations are based on evidence-based international clinical practice guidelines, clinical trial results, and systematic reviews. We have also adapted the diagnostic workup utilized in adults for children, taking into account that the results have not yet been validated in children.

## 2. Adverse Reactions to Iodinated Contrast Media

Different types of commercially available RCM are used in pediatric age. Iodinated contrast media (ICM) are used for computed tomography (CT) and fluoroscopic intervention (angiography, urography, cholangiography, hysterosalpingography). They are classified as water-soluble RCM (Table 1) and water-insoluble RCM. Ethiodized poppyseed oil is the only currently used water-insoluble ICM for embolectomy, sclerotherapy, and hysterosalpingography. Noniodinated RCMs in children include barium sulfate for gastrointestinal tract imaging.

Adverse reactions to RCM have been commonly distinguished in type A reactions (TAr) (toxicity and side effects) and type B (HRs or allergic-like) reactions [2,3]. TAr were previously known as physiologic reactions, and for these reasons, they have not been reported in many papers. However, it has recently been pointed out [4] that the term “physiologic” to define type A reactions is confusing and misleading, and it should not be used. Among all adverse reactions to RCM, the frequency of TAr is greater than 80% [5]. In a recent review [6], the frequency of HRs to ICM varied from 1% to 12%, with severe reactions (mainly anaphylaxis) accounting for 0.01–0.2% of total reactions. The frequency of HRs to ionic ICM is higher (3.8–12.7%) [7] than nonionic ICM (0.5–3%) [8]. The exact frequency of HRs is unknown in the pediatric population, but it is lower than in adults [9]. Katayama et al. [7] collected data from more than 100,000 administrations of ionic/nonionic CM. By stratifying the patients by age, use of ionic/nonionic CM, and severity of reactions, the authors showed that children (<10 years of age) and elderly patients (>70 years of age) have the lowest rates of severe adverse reactions, with an incidence of 0.07% both for ionic/nonionic CM in children aged 1–9 years, 0.41% for ionic CM, and 0.07% for nonionic CM in children aged 10–19 years. Fjelldal et al. [10] described 5/547 allergic-like reactions to iohexol in children, with a 0.9% reaction rate. Mikkonen et al. [11] found 1.9% of children with acute adverse reactions and 6.2% of children with late reactions using a parent-filled questionnaire. The reactions were classified according to Ansell [12], who included signs and symptoms typically considered to be TAr. Dillman et al. [13] reviewed more than 11,000 doses of nonionic low-osmolality ICM administered to children and neonates, recording a total of 20 (0.18% of the patients) acute allergic-like reactions. Most reactions were mild (80%), one (5%) was moderate, and three (15%) were severe. In particular, reactions to iohexol were 14/7963 doses, and to iopromide, they were 6/3343 doses. Three patients (0.027% of total) had severe reactions, but two out of three had already had an allergic-like reaction, suggesting that a certain predisposition could be claimed. In the same population, the reaction rate in adults was 0.6% [14], confirming a lower rate of HR in children. In 12,494 consecutive patients up to 21 years of age, Callahan et al. [9] observed an incidence of 0.46% of reactions (allergic-like + physiologic), the majority of which were mild (82%). In this study, nearly 70% of all mild CM reactions occurred in patients between 10 and 18 years, and nearly half occurred in adolescents. Furthermore, only a small percentage (5%) of patients with adverse reactions to contrast material had any prior documented history of asthma. Even patients with a history of asthma had moderate reactions. Dewachter et al. [15] reported a 13-year-old child with a grade 2 severity reaction (characterized by “moderate cutaneous-mucous, cardiovascular or respiratory signs”) to ioxitalamate. There are some factors that could increase the risk for HRs to ICM, including some allergic-related factors such as a previous history of drug allergy and atopy; some drug-related factors such as concomitant treatment with IL-2, ACE-inhibitors, beta-blockers, or proton pump inhibitors; some patient-related factors such as concomitant renal disease; cardiovascular disease; and gender (female sex prevalence/dominance) [16]. No specific risk factors have been identified in children. Dillman et al. found a female prevalence in children, as well as other possible risk factors such as a previous history of reaction to ICM, bronchial asthma, and previous allergic-like reaction to allergens other than ICM [13,17].

## 3. Clinical Presentation and Pathophysiology

HRs include immediate (<1 h) and nonimmediate (≥1 h to 10 days) reactions (Figure 1) [8,18]. According to the Gell and Coombs classification, nonimmediate allergic reactions consist of different endotypes: IgG cytotoxic and complement reactions or type 2, immune-complex, IgG-mediated complement reactions or type III, and T-cell-mediated reactions or type IV [19].

The severity of reactions could be classified according to the American College of Radiology [1]. Skin and mucous involvement characterized by erythema and urticaria with or without angioedema affects up to 70% of patients with reactions to ICM [16]. According to a recent review, severe reactions account for 0.01% to 0.2% of all reactions [6]. Only a small proportion of immediate reactions seems to be IgE mediated, but these data could change according to the studied population, being much higher in studies analyzing patients with severe reactions [18,20]. Positive results of skin tests and in vitro tests (increased serum tryptase levels, histamine release from basophils and mast cells, and basophil activation tests (BATs)) to ICM support an immediate underlying mechanism. In adults, the frequency of allergy increased when >2 organs were affected and when high histamine and tryptase levels were detected [21]. Most immediate reactions are not mediated by a hypersensitivity mechanism. Immediate reactions are thought to be triggered by various mechanisms, including complement activation, direct mediator release from basophil and mastocytes, activation of platelets and endothelium cells, and bradykinin involvement. Therefore, these types of adverse reactions can be defined as pseudoallergic instead of allergic since an underlying hypersensitivity mechanism is lacking, which could also explain why some patients experience severe reactions to ICM following first exposure. Nonimmediate HRs develop in 0.5% to 23% of patients [6]. These are commonly mild skin reactions (30–90% of cases) with mainly urticaria/angioedema and maculopapular exanthema such as skin eruptions [16]. Severe cutaneous adverse reactions (SCARs) (Stevens–Johnson syndrome, toxic epidermal necrolysis, acute generalized exanthematous pustulosis, drug reaction with eosinophilia and systemic symptoms, generalized bullous fixed-drug eruption), symmetric drug-related intertriginous, and flexural exanthema have been seldom reported, with two pediatric cases described (a 6-year-old child with Stevens–Johnson syndrome to iopentol and a 14-year-old patient with toxic epidermal necrolysis to ioxaglate and iohexol at 3 years of age and iopamidol at 12 years of age) [22,23]. Histological investigations showing T-cell perivascular infiltrates indicate that nonimmediate HRs are expected to be T cell mediated [24]. RCM could elicit reactions that resemble HRs [25,26,27,28,29]. As reported above, these reactions are classified as TAr, with mainly systemic symptoms such as vasovagal reactions, flushing, itching, warmth sensation, chills, tachycardia, and headaches. These reactions generally develop within 1 h and are more frequently observed during angiography due to the direct arterial administration of RCM at high doses. The treatment is supportive; nonetheless, patients presenting this minor manifestation should be monitored to prevent any pseudoallergic reaction, which may initially show similar prodromes [30,31]. TAr to ICM could be due to chemotoxicity, osmotoxicity, or direct toxicity of endogenous molecules. These reactions have been gradually decreased in incidence with the use of nonionic CM. The chemotoxicity of the ICM molecule is mostly related to the number of carboxyl and hydroxyl groups: whereas the chemotoxicity increases proportionally to the number of carboxyl groups, the hydroxyl groups decrease the chemotoxicity of the ICM molecule. The mechanisms underlying the chemotoxicity are not well understood, but they may be related to protein-binding affinity in plasma or cell membranes, dysfunctional activation of the complement, kinin, fibrinogen, and coagulation cascade, or release of vasoactive substances from cells. These mechanisms may also trigger pseudoallergic symptoms. Osmotoxicity is another characteristic of the ICM molecule to be considered. This feature is related to the hypertonicity of ICM solutions compared to the plasma, and it is linked to the ratio of the number of iodine atoms to the number of particles in the solution. This property could explain side effects such as pain when arteriography is performed or hypotension due to vasodilatation. In neonates and toddlers in particular, who have a lower ability to balance osmotic shifts, hyperosmolar intravascular contrast media could cause a fluid shift from the extravascular compartment to the bloodstream. Therefore, heart failure and pulmonary edema may occur in this group of young children [32,33]. Finally, the direct toxicity of the ICM molecule caused by direct interference with cellular functions has to be taken into account. This is due to a too-high or too-low concentration of ions. The four groups of water-soluble ICM greatly differ in their osmotic and chemotoxic properties: ionic monomers have the highest osmotoxicity, as well as the highest chemotoxicity, due to a large number of carboxyl groups; conversely, nonionic dimers are the most tolerated ICM due to the absence of carboxyl groups and have the lowest osmotoxicity.

## 4. Diagnostic Tests

A complete allergy workup is recommended in the case of adverse reactions to ICM [18]. HRs must be accurately ascertained to avoid overdiagnosis of allergy to ICM. It is also recommended to test different ICMs along with the culprit to identify an alternative drug with negative allergy tests for use in the case of future imaging procedures.

### 4.1. Skin Tests

A skin prick test (SPT) should be performed to identify patients with ICM-triggered HRs and, in the case of a negative intradermal test (IDT), should be performed together with a patch test in delayed reactions. Skin tests should be performed with the culprit agent or if the history is unclear, with the broadest panel of ICM. When the culprit is positive, a wide panel of RCM should be tested to identify a safe, alternative ICM for use in the future in the case of necessity [34,35]. Cross-reactivity between ICM has been frequently reported. Lerondeau et al. [36] showed positive skin tests to multiple ICMs in 67% of 97 patients with immediate and delayed HRs, independently from the chemical classification. The recommended concentrations for the SPT and IDT are reported in Table 2. IDT readings are performed after 20 min in immediate reactions and after 20 min, 48 h, and 72 h in nonimmediate reactions. The patch test should be removed after 48 h with a reading at 48 and 72 h. Patients should be advised to report any delayed reactions. If the patient observes a positive reaction at the skin test site at other time points, additional readings should be performed up to 7 days for the IDT and at 102 h for the patch test [20].

In adults with a diagnosis of immediate HRs to ICM confirmed by a provocation test or re-exposure to the culprit ICM, the positive rate of skin tests (SPT and/or IDT) ranged between 5.6 and 64.7% [6]. This discrepancy may be explained by several factors, including skin test technique, type of ICM, severity of reaction, and interval time between reaction and tests. Patients with a history of immediate HRs had positive ICM skin tests in approximately 17% of cases and in about 52% of severe reactions [37]. Skin tests should be performed between 6 weeks and 6 months after the acute reaction to obtain the highest rate of the positive test [18]. In patients with a history of immediate reaction to ICM, positive SPTs were observed in 3% of cases and positive IDTs in 25% of cases at 20 min and in 2.5% at 10–24 h [18]. The specificity of the SPT was found to be 94.6%, and for the IDT, it varied from 91.4% to 96.3% [18,38]. The negative predictive value of the skin test to ICM was high, around 96% in 64 patients who were exposed again to the ICM [6]. On the other hand, there is a paucity of data on the positive predictive value of skin test results. In seven patients with a positive IDT to the culprit ICM, re-exposure elicited a reaction in five cases [39,40]. In patients with nonimmediate reactions, an SPT, late IDT readings, and a patch test with readings at 48 and 72 h [41] should be performed [34]. When the IDT with 1:10 dilution is negative, the undiluted ICM can be tested. In 10 studies, SPTs, IDTs, or patch tests were positive in 16.9–53.3% of cases, the negative predictive value varied from 0 to 100%, and the positive predictive value varied from 50% to 100% [6]. In a large study, Brockow et al. [18] found that skin tests were positive in 37 out of 98 nonimmediate reactors (38%), SPTs in 3% of cases, delayed IDTs in 32% of cases, and patch tests in 28% of cases. The negative predictive value of skin tests for nonimmediate reactions has been shown to be low, so the diagnosis often requires the drug provocation test (DPT). Patients with a negative IDT and patch tests reacted on re-exposure to the suspected ICM in 17 (41.67%) cases [42,43]. However, Schrijvers et al. [35] observed that only 6 out of 34 (17.6%) patients with positive skin tests had positive challenges, and the negative predictive value was 86.1%. In fixed-drug eruption, a delayed reaction can be observed when the IDT and the patch test are performed on the site of the eruption but not in different areas, probably due to the persistent localized memory of the T cell to the ICM [44]. In SCARs, it is advisable to perform IDTs when patch tests are negative.

### 4.2. Tryptase

In doubtful cases, it could be useful to perform a blood draw for tryptase dosage to be further compared with basal tryptase level, although the role of elevated tryptase has not yet been studied in ICM HRs. As a matter of fact, serum levels of histamine and tryptase have been found to be elevated only in a few patients who have experienced a severe reaction to ICM. Furthermore, in patients who have experienced mild symptoms, no elevation of tryptase or histamine has been found. Therefore, these markers have yet to be proven to be a useful diagnostic tool. Nonetheless, blood sampling for tryptase has been recommended [45]. An increase of at least 2 ng/mL + (1.2 × baseline tryptase level) or at least 20% above baseline plus 2 ng/mL within 4 h after the reaction over baseline tryptase levels suggests an immediate HR to ICM [20]. It is necessary to determine the baseline value since it can be elevated in hypertryptasemia and in mast-cell diseases. Since there is a lack of reliable diagnostic means of HRs to ICM in real time, it would be of interest to investigate whether exhaled breath biomarkers can be helpful to elucidate the mechanisms of the reaction [46,47,48].

### 4.3. Basophil Activation Test

BAT detects specific markers of activated basophils by incubation with ICM. Sensitivity ranges from 46% to 62%, while specificity seems to be between 88% and 100% [16,34,49,50]. The test needs to be validated in larger samples.

### 4.4. Other “In Vitro” Tests

The lymphocyte activation test seems to be a promising test to identify delayed HRs. It identifies specific memory T cells proliferating after stimulation with the culprit drug. The test should be performed between 4–8 weeks and 2–3 years from the reaction [51,52]. However, sensitivity greatly varies from 13% to 75%. Sensitivity has not yet been established, and reactions have also occurred in patients with negative results [20]. Therefore, the test is currently a research tool. Some investigators have reported the presence of serum ICM-specific IgE antibodies in some patients with immediate reaction to ICM [53,54]. Despite this, there has been great variability in the percentage of positive patients, so the usefulness of such a test is still to be established. Since direct histamine release by leukocytes could be stimulated by ICM in a dose-dependent fashion, the histamine release test has been performed in some cases. It has been described that leukocytes from atopic individuals are more prone to release higher quantities of histamine upon CM stimulation than leukocytes from nonatopic patients. A similar difference has been found between patients with an adverse reaction and those without a previous reaction: the former has considerably higher levels of histamine released. As for the other tests mentioned, the role of the histamine release test has yet to be defined [8].

## 5. Management

The DPT is the gold standard for diagnosing HR to RCM. However, the DPT is a risky, time-consuming procedure that needs to be supervised by trained personnel with equipment and support in the case of severe reactions. Several protocols have been used. Generally, it should not be administered in more than four graded doses to avoid desensitization. The starting step should be 1/10 of the total dose, and the interval time between each step should be 1 h [55]. Blood pressure should be monitored [56]. It is important to underline that the administration of low doses of ICM before the DPT is not recommended since even minimal “pretest” applications could lead to severe reactions [57]. Therefore, the DPT should be performed after a careful assessment of the risk–benefit balance. The main reason for performing the DPT is to confirm whether an alternate ICM negative to the skin test safe in patients with a positive skin test and/or severe reaction (anaphylaxis) to the culprit ICM [20,34]. In patients with a history of mild immediate (urticaria/angioedema) or nonimmediate reactions (maculopapular exanthema) and negative skin tests to the culprit ICM, a diagnostic DPT with the culprit ICM may be considered to reach a confident diagnosis [20]. However, in these cases, the use of an alternative ICM negative to the skin test should be the preferred option when possible. For practical reasons, re-exposure to ICM during radiological examinations can replace the DPT. When patients with immediate skin reactions do not undergo skin tests, ICM can be administered following premedication, and an alternative ICM should be administered if the ICM is not known. In the case of a previous anaphylactic reaction, native CT and magnetic resonance should be used [20]. RCM is contraindicated in patients with SCARs who should undergo native CT and magnetic resonance. In patients with immediate reactions, acute desensitization is a feasible option in the case of urgent/emergency use of ICM without testing sensitization to available agents [58]. Desensitization has been shown to be effective in adults [58], and it may be convenient in children too. After pretreatment with prednisone, diphenhydramine, and ranitidine, a 13-step desensitization protocol starting with a 10,000-fold dilution of the full dose administered intravenously and doubled every 10 min has been proposed [59]. The role of premedication is still debated. There is some evidence that premedication can be useful in patients with previous immediate reactions [60]. However, it seems that premedication prevents mild HRs, but not severe HRs or nonimmediate HRs [61]. The premedication protocol used in adults with 50 mg of prednisone orally at 13, 7, and 1 h before ICM administration and 1 mg/kg diphenhydramine IM or orally 1 h before [34] has been modified by Lindsay et al. [62]. In children, it has been adapted as follows: 0.5 mg/kg methylprednisolone orally or IV at 13, 7, and 1 h before ICM administration, and 1 mg of chlorphenamine at 1–5 years, 2 mg at 6–12 years, and 4 mg orally over 12 years 1 h before ICM administration. Keeping in mind that some patients react to multiple CMs, another question is whether a safe CM can be identified without performing allergy testing or the DPT. Unfortunately, so far, this is not feasible [63]. On the one hand, an exact grouping method to classify the chemical groups of ICMs is lacking. On the other hand, there are cross-reactions not only between compounds belonging to different chemical groups but also between those within the same chemical group, even if they are less frequent. That being said, a clinical decision should consider that individual predisposition to react to several CMs is more important than similarities in chemical structure [63].

## 6. Conclusions

Adverse reactions to CM in pediatric age seem to be rare, and this is reflected in the paucity of studies conducted in children. Overall, these studies, albeit with small differences between the incidence rates, probably due to the different classification of side effects such as adverse reactions, confirm that the use of CM in pediatric radiology is quite safe in terms of frequency of reaction occurrence, as well as in terms of incidence of severe events. They also highlight the existence of risk factors such as previous reactions to RCM, bronchial asthma, and previous allergic-like reactions to substances other than contrast material. Despite the lack of desired pediatric guidelines, management employed in adults could be translated into the pediatric setting, taking into account that the results have not yet been validated in children. An IgE-mediated mechanism is responsible only for a minority of immediate reactions. However, the IDT and SPT are recommended for risk stratification and to find safe alternatives [20,34]. More studies are warranted to understand whether DPT should be routinely performed in the allergy workup.

## Figures and Tables

**Figure 1 medicina-58-00517-f001:**
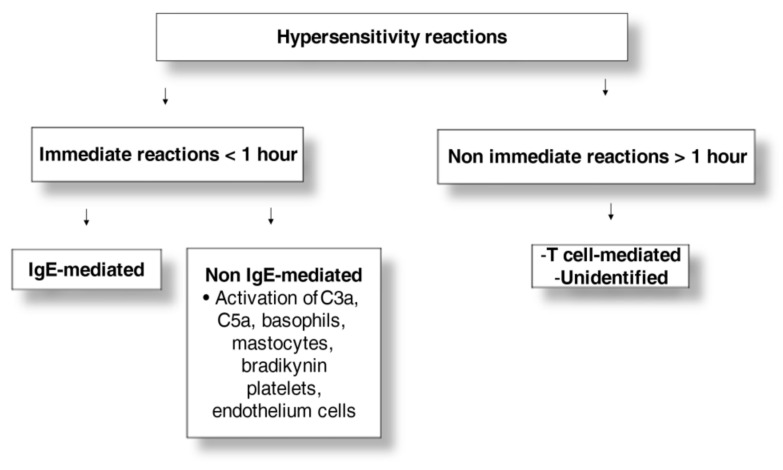
Hypersensitivity reactions to radiocontrast media.

**Table 1 medicina-58-00517-t001:** Classification of water-soluble iodinated contrast media.

Class	Combination		Iodine Content (mg/mL)	Osmolality (mOsm/kg)
Ionic monomers with high osmolality	Meglumine iothalamate Na	Conray^®^	325	1843
	Meglumine diatrizoate Na	Gastrografin^®^	306	1530
	Metrizoate Na	Isopaque^®^	370	2100
Ionic dimers with low osmolality	Ioxaglate acid	Hexabrix^®^	320	580
	Iodipamide	Cholografin^®^ Meglumine^®^	260	
	Iotroxate	Biliscopin^®^	105	600
Nonionic monomers	Iopamidol	Iopamiro^®^	300	616
	Iohexol	Omnipaque^®^	300	640
	Ioversol	Optiray^®^	320	702
	Iopentol	Imagopaque^®^	250	350
	Iomeprol	Iomeron^®^	400	726
	Iopromide	Ultravist^®^	300	590
	Iobitridol	Xenetix^®^	350	915
	Ioxilan	Oxilan^®^	350	721
Non ionic dimers	Iotrolan	Isovist^®^	300	320
	Iosimenol *		340	290
	Iodixanol (isoosmolal)	Visipaque^®^	320	290

* Not commercially available.

**Table 2 medicina-58-00517-t002:** Recommended concentrations for skin tests with ICM.

Test	Concentration
Skin prick test	1:1
Intradermal test	1:10 1:1 only in nonimmediate reaction
Patch test	1:1 only in nonimmediate reaction

## Data Availability

Not applicable.

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
