# Peer review of "Radiocontrast Media Hypersensitivity Reactions in Children"

_medicina, 2022, doi:10.3390/medicina58040517_

Round 1
Reviewer 1 Report
This manuscript is a narrative review of hypersensitivity reactions to RCM in children. The topic has clear practical significance, is of general interest and remains as an unsolved clinical problem; therefore, such an article deserves publishing. However, there is a number of issues that have to be resolved before acceptance:
- the review is narrative, so the reader cannot be sure that the authors have reviewed all relevant literature. I suggest to the authors to add short methodology section explaining at least what databases were searched, what was the search strategy and what were the inclusion and exclusion criteria
- the manuscript is dealing with serious clinical problem, and clinicians are frequently in a dilemma what CM to choose when there was a history of allergy to CM or other drugs; the authors have given some recommendations, but we have to know on what grounds. Therefore, for key clinical recommendations the authors should shortly analyze quality of evidence (what type of studies were made, what was the risk of bias, etc.) they based their recommendations on.
- Some of the CM were not mentioned in the table - please include all that are widely available: e.g. ioxilan, iobitridol, etc.
- a paragraph should be devoted to cross-allergy between the RCM;
- please do not call type A adverse drug reactions as "physiological". These are the reactions that reflect extension of main mechanism of action of a drug, but they are by no means physiological
- please divide nonimmediate reactions to type 2, 3 and 4 allergic reactions
Author Response
Reviewer 1
This manuscript is a narrative review of hypersensitivity reactions to RCM in children. The topic has clear practical significance, is of general interest and remains as an unsolved clinical problem; therefore, such an article deserves publishing. However, there is a number of issues that have to be resolved before acceptance:
- the review is narrative, so the reader cannot be sure that the authors have reviewed all relevant literature. I suggest to the authors to add short methodology section explaining at least what databases were searched, what was the search strategy and what were the inclusion and exclusion criteria
Answer:
We thank the reviewer for his/her criticism. Accordingly, the requested information have been given in the introduction.
- the manuscript is dealing with serious clinical problem, and clinicians are frequently in a dilemma what CM to choose when there was a history of allergy to CM or other drugs; the authors have given some recommendations, but we have to know on what grounds. Therefore, for key clinical recommendations the authors should shortly analyze quality of evidence (what type of studies were made, what was the risk of bias, etc.) they based their recommendations on.
Answer: We are are grateful for your comment. We have added to the text that due to the paucity of guidelines and studies on HRs to CM in pediatric age, it is difficult to apply current system for quality of evidence (as GRADE system). Most of our recommendations are based on evidence-based international clinical practice guidelines, clinical trials results, and systematic reviews.
- Some of the CM were not mentioned in the table - please include all that are widely available: e.g. ioxilan, iobitridol, etc.
Answer: As requested, an extended list of CMs has been provided in Table 1.
4.a paragraph should be devoted to cross-allergy between the RCM;
Answer: We thank the referee for the opportunity to better address this point. Unfortunately, a correct grouping method to classify chemical groups of ICMs is lacking. Furthermore, patients with HR to a compound can have reaction to a different compound, irrespective to the chemical group it belongs. That being said, individual predisposition to react to several CMs is more important than similarities in chemical structure. A paragraph on this issue has been added at the end of the secrtion.
5.please do not call type A adverse drug reactions as "physiological". These are the reactions that reflect extension of main mechanism of action of a drug, but they are by no means physiological
Answer: We understand the reviewer’s point of view. The sentence has been corrected as follow: Adverse reactions to RCM are distinguished in HRs (or allergy-like) and type A reactions (toxicity and side affects). References have been added.
6.please divide nonimmediate reactions to type 2, 3 and 4 allergic reactions
Answer: As requested, we have detailed nonimmediate reactions in the text.
Reviewer 2 Report
This paper is good and the content is well presented and organized.
Author Response
This paper is good and the content is well presented and organized.
Answer: We sincerely thanks the reviewer for the positive comment on our study.
Reviewer 3 Report
MINOR CORRECTIONS AND SUGGESTIONS:
In the section title "2. Adverse reactions to ICM" one should NOT use that abbreviation but the whole syntagm: "Iodinated contrast media" (with the abbreviation defined in the paragraph).
Given the relatively large number of abbreviations, I recommend to authors to create a table of abbreviations at the start of this paper (immediately after the abstract): it would be even more efficient to also save this abbreviation table as a separate pdf to be downloaded (from a separate URL that should be mention in the caption of that table inside the main paper) and to be opened at the same time with the main article (so that to easily re-call a specific abbreviation when reading the main paper). Another additional suggestion would be to put those abbreviations in bold (under the same parentheses) so that to be easily observed when the reader reads a subsequent paragraph and wants to re-call a specific abbreviation (defined in a previous paragraph).
I suggest using “contrast media” (instead of the “CM” abbrev.) in the caption of Table 1: it is generally better for the captions (of tables and images) to not contain abbreviations, so that to be easily readable even when extracted from the main paper.
I suggest the abbreviation “mL” instead of “ml” (in “mg/ml”) in the 1st table. The text “(mOsm/kg)” should appear on the same line (in the same table 1). The Ⓡ symbol should appear as superscript in the 1st table (and in general) (with a smaller font) and on the same line as the commercial name of that substance.
Please reformulate and clarify this phrase (from line[L] 53): “Type A reactions are about 80% of the total number of RCM adverse reactions (to both ionic and non-ionic RCMs?)”
L51-52: I suggest to also abbreviate the “physiologic (or Type A) reactions” as “TARs” and to give some examples of HRs and TARs under parentheses.
I suggest using (and keeping) the term “frequency” instead of “incidence” (because frequency is expressed in percents, but incidence is expressed in various other units of measure as a function of number of persons/patients and time interval) in concordance with the paragraph L51-52 (in which the term “frequency” is used), as “frequency” and “incidence” are not exact synonyms, but quasi-synonyms.
TYPOS AND SUGGESTIONS:
-- “but lower than in adults” (L55), “authors” instead of “Authors” (L57);
-- define the age of those “elderly patients” (L58)
-- define “mild” (L60) between parentheses (given some examples of such “mild” HRs and Type-A reactions respectively)
-- define “reaction rate” (L61): does “reaction” means both HRs and Type-A reactions (summed)?
-- define “reaction rate” (L62): does “reaction” means both HRs and Type-A reactions (summed)?
At L63 (“Three patients have showed severe reactions”) also put between parentheses what percent does those 3 patients represent from the total of analyzed patients (so that to can compare with the given 0.6% from L66).
At L68 please give the percent so that to defined what exactly “majority” means in this specific study, because “majority” is a vague term that could mean anything between 40-50% and 99%.
At L70 please define exactly that “small percent”.
At L73, define “a grade 2 severity reaction” under parentheses.
At L79 use “female sex prevalence/dominance” instead.
L80 contains an unclear expression: “as per other possible risk factors […]” what? in those risk factor does the female sex predominate? (if so, do not put the period sign after “significance”).
L83: missing parenthesis for “< 1 hour)” and use the sign “≥” in the formula “(≥ 1 hour to 10 days)” (and in the Figure 1)
L88: “70% of cases” of all types of HRs triggered by ICMs? What is the overall frequency of all types of HRs for ICMs? Does it refer to children only? I ask because 70% is a very high frequency.
L89: “More severe reactions (including vomiting, abdominal pain …)”, but with what frequency?
L90: “Only a small part”: please give an approximate percent.
L96-99: But complement activation, direct mediator release from basophils and mastocytes etc are also immunologic mechanisms: why to call them “pseudo-allergic instead of allergic”. May be the author means that they aren’t actually hypersensitivity mechanisms (a specific subtype of immunologic mechanisms).
L102: I suggest to replace “(30-90%)“ with:: “(in 30-90% of cases)”
L107: Define “seldom” as an approximate percent.
L109: Use the “TAR(s)” abbreviation for “type A (physiological) reactions” (as also previously proposed)
L115-116: Then use “ICM-triggered TARs” instead of “Type A reactions to ICM”
L117: Use “decreased in incidence” instead of “decreased in number”.
L131: Use “extravascular compartment” instead of “extravascular tissue”.
L132: Use “may happen” instead of “could happen”.
L134: Use “caused by the […]” instead of “caused to the […]”.
L134: Use “to a too high [….]” instead of “to the too high […]”
L137: Use “large number” instead of “high number”.
L144: Use “alternative ICM” instead of “alternative one”
L147: Use “ICM-triggered HR” instead of “HR to ICM”
In Table2, the 2nd column should be named “Dilution ratio” (and a solvent of dilution like NaCl 0.9% should be mentioned). “Positive test” criteria for those 3 types of skin tests should also be mentioned.
L176-177: give the reference of that specific study at the end of the proposition.
FINAL REMARK. In my opinion, any review process may also contain a dialog between the reviewer/editor and the authors (not only reviewer’s/editor’s monologue) and that is why any of my suggested corrections and suggestions are debatable, so that the author(s) can send me a counter argument if he/she does not agree to any of my suggestions / critique and he/she wants to bring more clarifications to the issues invoked by me as a reviewer of this paper.
Author Response
MINOR CORRECTIONS AND SUGGESTIONS:
In the section title "2. Adverse reactions to ICM" one should NOT use that abbreviation but the whole syntagm: "Iodinated contrast media" (with the abbreviation defined in the paragraph).
Answer: As requested, the paper has been corrected
Given the relatively large number of abbreviations, I recommend to authors to): it would be even more efficient to also save this abbreviation table as a separate pdf to be downloaded (from a separate URL that should be mention in the caption of that table inside the main paper) and to be opened at the same time with the main article (so that to easily re-call a specific abbreviation when reading the main paper). Another additional suggestion would be to put those abbreviations in bold (under the same parentheses) so that to be easily observed when the reader reads a subsequent paragraph and wants to re-call a specific abbreviation (defined in a previous paragraph).
Answer: As requested, we have inserted a table of abbreviations after the abstract.
I suggest using “contrast media” (instead of the “CM” abbrev.) in the caption of Table 1: it is generally better for the captions (of tables and images) to not contain abbreviations, so that to be easily readable even when extracted from the main paper.
Answer: The caption has been modified.
I suggest the abbreviation “mL” instead of “ml” (in “mg/ml”) in the 1st table. The text “(mOsm/kg)” should appear on the same line (in the same table 1). The Ⓡ symbol should appear as superscript in the 1st table (and in general) (with a smaller font) and on the same line as the commercial name of that substance.
Answer: Table 1 has been modified in agreement with suggestions.
Please reformulate and clarify this phrase (from line[L] 53): “Type A reactions are about 80% of the total number of RCM adverse reactions (to both ionic and non-ionic RCMs?)”
Answer: This sentence has been moved and has been modified.
L51-52: I suggest to also abbreviate the “physiologic (or Type A) reactions” as “TARs” and to give some examples of HRs and TARs under parentheses.
Answer: As requested, “Physiologic” has been erased and “TARs” has been used. A brief description of Type A reactions has been added to the text.
I suggest using (and keeping) the term “frequency” instead of “incidence” (because frequency is expressed in percents, but incidence is expressed in various other units of measure as a function of number of persons/patients and time interval) in concordance with the paragraph L51-52 (in which the term “frequency” is used), as “frequency” and “incidence” are not exact synonyms, but quasi-synonyms.
Answer: “incidence” has been changed for “frequency” throughout the text.
TYPOS AND SUGGESTIONS:
-- “but lower than in adults” (L55), “authors” instead of “Authors” (L57);
Answer: The word has been modified.
-- define the age of those “elderly patients” (L58)
Answer: The sentence has been modified.
-- define “mild” (L60) between parentheses (given some examples of such “mild” HRs and Type-A reactions respectively)
Answer: The line has been removed, since the previous sentence has been rewritten.
-- define “reaction rate” (L61): does “reaction” means both HRs and Type-A reactions (summed)?
Answer: In this paper only a clinical description of observed signs and symptoms has been reported as follow: “a mild periorbital and facial oedema in a girl 3 days of age; one 2-year-old girl devel- oped minor urticaria and started sneezing; one 4-year-old girl got urticaria on her neck, shoulder and back; and one 13-year-old girl with known allergy started sneezing. These four reactions came immediately after injection. Widespread urticaria developed 20 min after injection in a boy of 3 years”. No allergic tests have been performed to identify the nature of these reactions.
-- define “reaction rate” (L62): does “reaction” means both HRs and Type-A reactions (summed)?
Answer: as reported immediately after, these reactions are allergic-like. The sentence has been modified, showing data according to patients and doses. In the paper Authors specified that physiologic reactions had been considered as side-effect and were not documented.
At L63 (“Three patients have showed severe reactions”) also put between parentheses what percent does those 3 patients represent from the total of analyzed patients (so that to can compare with the given 0.6% from L66).
Answer: Percentages would be 0.027 (added to the text), but the 0.6% of adult population reaction rate referred to all reactions. In the adult population, 77% reactions were mild, 21% were moderate, and 2% were severe. In children, reactions were mild in 80%, moderate in 5% and severe in 15%.
At L68 please give the percent so that to defined what exactly “majority” means in this specific study, because “majority” is a vague term that could mean anything between 40-50% and 99%.
Answer: The percentage (82%) has been given.
At L70 please define exactly that “small percent”.
Answer: The percentage (5%) has been given.
At L73, define “a grade 2 severity reaction” under parentheses.
Answer: In the paper, Authors considered a grade 2 reactions as follow: Grade II corresponds to moderate clinical signs associating cutaneous-mucous, cardiovascular or respiratory signs”. This has been added to the text.
At L79 use “female sex prevalence/dominance” instead.
Answer: This has been corrected in the text.
L80 contains an unclear expression: “as per other possible risk factors […]” what? in those risk factor does the female sex predominate? (if so, do not put the period sign after “significance”).
Answer: The meaning is that there is not statistical significance also for other possible risk factors (like history of reaction to ICM, bronchial asthma and previous allergic-like reaction to allergens other than ICM), as reported for female gender. For greater clarity, "per" has been replaced with "as well as".
L83: missing parenthesis for “< 1 hour)” and use the sign “≥” in the formula “(≥ 1 hour to 10 days)” (and in the Figure 1)
Answer: The text has been corrected according to the reviewer’s comment.
Round 2
Reviewer 1 Report
I have read the revised manuscript carefully and I think that the manuscript has been sufficiently improved to warrant publication in Medicina.Author Response
Thank you for your appreciation. English language has been revised throughout the manuscript.
